# The Clinical Utility of Serum Alpha-1-Acid Glycoprotein in Reflecting the Cross-Sectional Activity of Antineutrophil Cytoplasmic Antibody-Associated Vasculitis: A Single-Centre Retrospective Study

**DOI:** 10.3390/medicina60081212

**Published:** 2024-07-26

**Authors:** Juyoung Yoo, Taejun Yoon, Yong-Beom Park, Sung Soo Ahn, Sang-Won Lee

**Affiliations:** 1Division of Rheumatology, Department of Internal Medicine, Yonsei University College of Medicine, Seoul 03722, Republic of Korea; gloriajuyoung@gmail.com (J.Y.); yongbpark@yuhs.ac (Y.-B.P.); 2Department of Medical Science, BK21 Plus Project, Yonsei University College of Medicine, Seoul 03722, Republic of Korea; tjyoonn92@gmail.com; 3Institute for Immunology and Immunological Diseases, Yonsei University College of Medicine, Seoul 03722, Republic of Korea; 4Division of Rheumatology, Department of Internal Medicine, Yongin Severance Hospital, Yonsei University College of Medicine, Yongin 16995, Gyeonggi-do, Republic of Korea

**Keywords:** alpha-1-acid glycoprotein, reflect, activity, antineutrophil cytoplasmic antibody, vasculitis

## Abstract

*Background and Objectives*: This study investigated whether serum alpha-1-acid glycoprotein (AGP) at diagnosis could reflect the cross-sectional activity represented by the Birmingham vasculitis activity score (BVAS) and further predict poor outcomes during follow-up in patients with antineutrophil cytoplasmic antibody-associated vasculitis (AAV). *Materials and Methods*: This study included 70 patients with AAV. Clinical data at diagnosis, including AAV-specific indices and acute-phase reactants such as erythrocyte sedimentation rate (ESR) and C-reactive protein (CRP), were reviewed. All-cause mortality, relapse, end-stage kidney disease (ESKD), cerebrovascular accident, and acute coronary syndrome were evaluated as poor outcomes of AAV. Serum AGP was measured using the sera obtained and stored at diagnosis. *Results*: The median age of the patients was 63.0 years, with 29 male and 41 female patients. The median serum AGP was 150.9 μg/mL. At diagnosis, serum AGP was significantly correlated with BVAS and ESR but not CRP or serum albumin. Additionally, serum AGP showed significant correlations with the sum scores of ear–nose–throat and pulmonary manifestations; however, no significant differences in serum AGP according to each poor outcome were observed. Although serum AGP at diagnosis tended to be associated with ESKD occurrence during follow-up, serum AGP at AAV diagnosis was not significantly useful in predicting the future occurrence of poor outcomes of AAV during follow-up. *Conclusions*: In this study, we demonstrated the clinical utility of serum AGP at AAV diagnosis in assessing the cross-sectional activity represented by BVAS in patients with AAV for the first time.

## 1. Introduction

Antineutrophil cytoplasmic antibody (ANCA)-associated vasculitis (AAV) is a small vessel vasculitis characterised by fibrinoid necrotising inflammation with no or few immune deposits in capillaries, arterioles, venules, and occasionally medium-sized arteries [1,2]. AAV consists of three subtypes including microscopic polyangiitis (MPA), granulomatosis with polyangiitis (GPA), and eosinophilic GPA (EGPA) based on clinical features: MPA predominantly provokes vasculitis in the lungs and kidneys; GPA often affects the upper and lower respiratory tracts; and EGPA exhibits clinical manifestations related to both/either allergic and/or vasculitic components [3,4,5,6]. To date, in addition to conventional inflammatory markers including erythrocyte sedimentation rate (ESR) and C-reactive protein (CRP), numerous efforts have been made to identify various serum biomarkers that can simultaneously estimate and reflect the cross-sectional activity of AAV assessed using the Birmingham vasculitis activity score (BVAS) as well as the poor outcomes of AAV, for instance, all-cause mortality or kidney function impairment requiring renal replacement therapy, predicted by the five-factor score (FFS). In particular, if the BVAS could be replaced with newly developed serum biomarkers, they may have advantages, such as convenience, reproducibility–reliability, and dynamic information compared to traditional AAV-specific indices [7,8,9].

Alpha-1-acid glycoprotein (AGP) is a negatively charged glycosylated single-chain protein, mainly synthesised and catalysed in the liver [10]. Similar to serum albumin, AGP is an essential plasma-binding protein for exogenous and endogenous substances, which may affect the pharmacokinetics of drugs as well as modulate immune responses [11]. Moreover, AGP has been known to act as an acute-phase reactant because its serum concentration may rise two to five times during the acute phase of inflammation [11,12]. Additionally, AGP has been investigated and proved to be a predictor of all-cause mortality comparable to traditional biomarkers [13]. Therefore, it could be reasonably assumed that serum AGP could reflect the cross-sectional activity and predict poor outcomes during the disease course in patients with AAV. Nonetheless, no study has evaluated the clinical usefulness of serum AGP in patients with AAV to date. Hence, in the present study, we investigated whether serum AGP at AAV diagnosis could reflect the cross-sectional activity represented by the BVAS and further predict poor outcomes during follow-up in patients with AAV.

## 2. Materials and Methods

### 2.1. Patients

In this single-centre retrospective study, we randomly selected 70 patients from the Severance Hospital ANCA-associated VasculitidEs (SHAVE) cohort, an observational cohort of Korean patients with AAV from November 2005 until the present study. The electronic medical records of the 70 patients were retrospectively reviewed. The inclusion criteria were (i) diagnosis of AAV according to the algorithms for AAV and polyarteritis nodosa proposed by the European Medicines Agency in 2007 and the revised International Chapel Hill Consensus Conference Nomenclature of Vasculitides suggested in 2012 [1,2]; (ii) fulfilment of the 2022 American College of Rheumatology and European Alliance of Associations (ACR/EULAR) for Rheumatology classification criteria for MPA, GPA, and EGPA [3,4,5]; (iii) the first diagnosis of AAV made at the Division of Rheumatology, Department of Internal Medicine, Yonsei University College of Medicine, Severance Hospital by two rheumatologists; (iv) presence of medical records sufficient for not only collecting clinical, laboratory, radiological, and histological data but also for classifying, assessing AAV activity at AAV diagnosis and confirming poor outcomes during the follow-up duration until the last visit or the occurrence of each poor outcome; (v) follow-up duration for 6 months or greater after AAV diagnosis; and (vi) patients whose sera were collected and stored on consent at AAV diagnosis. The exclusion criteria were (i) patients who had serious medical conditions such as concomitant malignancies and severe infectious diseases requiring hospitalisation at AAV diagnosis; (ii) patients who were diagnosed with overlap syndrome with other types of systemic vasculitis at AAV diagnosis; and (iii) patients who were exposed to immunosuppressive drugs for AAV treatment within four weeks before AAV diagnosis. This study followed the STROBE statements (Appendix A).

The present study was approved by the Institutional Review Board (IRB) of Severance Hospital, Seoul, Republic of Korea (IRB number 4-2016-0901), and conducted in accordance with the Declaration of Helsinki. All patients in this study provided written informed consent upon enrolment in the SHAVE cohort at the time of AAV diagnosis and blood sampling. The IRB waived the requirements for additional written informed consent when it was obtained upon enrolment into the SHAVE cohort.

### 2.2. Clinical Data

Variables at AAV diagnosis, including age, sex, smoking history, and body mass index, were collected as demographic data. AAV subtype, ANCA positivity, and AAV-specific indices, including BVAS (version 3) [8] and FFS [9], were reviewed. Perinuclear (P)-ANCA and cytoplasmic (C)-ANCA were detected by an indirect immunofluorescence assay, whereas myeloperoxidase (MPO)-ANCA and proteinase 3 (PR3)-ANCA were measured by an immunoassay, the novel anchor-coated and highly sensitive Phadia EliA (Thermo Fisher Scientific/Phadia, Freiburg, Germany) using human native antigens, performed on a Phadia 250 analyser. According to the 2022 ACR/EULAR criteria for AAV, in the present study, both MPO-ANCA/PR3-ANCA and P-ANCA/C-ANCA were recognised as ANCA results [3,4,5]. Comorbidities included type 2 diabetes mellitus, hypertension, and dyslipidaemia. Acute-phase reactants including ESR and CRP (reference range from 0 to 8 mg/L) were recorded. The results of routinely performed laboratory tests, such as white blood cell count, haemoglobin, platelet count, blood urea nitrogen, serum creatinine, total serum protein, and serum albumin, were also collected. During follow-up, all-cause mortality, relapse, end-stage kidney disease (ESKD), cerebrovascular accident (CVA), and acute coronary syndrome (ACS), which occurred after AAV diagnosis, were evaluated as poor outcomes of AAV. The follow-up duration based on each poor outcome was defined as the period from AAV diagnosis to the occurrence of each poor outcome for patients with that poor outcome, and as that from AAV diagnosis to the last visit for those without such an outcome. Additionally, the frequency of the administration of immunosuppressive drugs was also investigated.

### 2.3. Measurement of AGP

After obtaining written informed consent, whole blood was collected from patients with AAV. Sera were immediately isolated from whole blood and stored at −80 °C. According to the manufacturer’s instructions, the serum AGP levels were measured using an enzyme-linked immunosorbent assay kit (R&D Systems, Minneapolis, MN, USA).

### 2.4. Statistical Analyses

All statistical analyses were performed using SPSS version 26 (IBM Corporation, Armonk, NY, USA) for Windows (Microsoft Corporation, Redmond, WA, USA). Continuous and categorical variables are expressed as medians (25–75 percentiles) and numbers (percentages), respectively. Correlation coefficients (r) between two variables were determined using Pearson correlation analysis. Significant differences between two continuous variables were compared using the Mann–Whitney U test. Using receiver operating characteristic (ROC) curve analysis, a significant area under the curve (AUC) was determined. A Cox proportional hazard model was performed to obtain the hazard ratio (HR) for each poor outcome during follow-up. For supplementary data, the cut-off was extrapolated by performing ROC curve analysis and selected as the value with the maximum sum of sensitivity and specificity; a comparison of the cumulative survival rates between the two groups was performed using Kaplan–Meier survival analysis with the log-rank test. A *p*-value < 0.05 was considered statistically significant.

## 3. Results

### 3.1. Characteristics

The median age of the study participants at diagnosis was 63.0 years, with 29 and 41 men and women, respectively. In addition, 35, 20, and 15 patients were diagnosed with MPA, GPA, and EGPA, respectively. Myeloperoxidase (MPO)-ANCA (or perinuclear [P]-ANCA) and proteinase 3 (PR3)-ANCA (or cytoplasmic [C]-ANCA) were detected in 40 (57.1%) and 11 (15.7%) patients, respectively. The median BVAS, FFS, ESR, and CRP were 5.0, 0, 23.0 mm/h, and 3.4 mg/L, respectively. The median serum AGP was 150.9 μg/mL. During follow-up, 4 patients died of any cause, and 11 experienced relapses. Additionally, fifteen, four, and one had ESKD, CVA, and ACS, respectively. Sixty-nine (98.6%) patients received glucocorticoid therapy, and the most frequently administered immunosuppressive drug was cyclophosphamide (64.3%), followed by azathioprine (60.0%) (Table 1).

### 3.2. Correlation Analysis

Serum AGP was significantly correlated with the BVAS (r = 0.255, *p* = 0.033) and ESR (r = 0.031, *p* = 0.015) at AAV diagnosis. However, no significant correlation was observed between serum AGP and CRP or serum albumin (Figure 1). Additionally, among the nine systemic items of the BVAS, serum AGP showed significant correlations with the sum scores of both ear–nose–throat (ENT) and pulmonary manifestations (r = 0.388, *p* = 0.001, and r = 0.238, *p* = 0.048, respectively) (Figure 2A). However, when comparing serum AGP according to the presence of ENT or pulmonary manifestation, no significant differences were observed in the serum AGP between the two groups (Figure 2B).

### 3.3. AUC of Serum AGP for Poor Outcomes

We performed ROC curve analyses to assess the predictive ability of serum AGP for poor outcomes. We, however, found no significant AUCs of serum AGP at diagnosis for all-cause mortality, relapse, ESKD, CVA, and ACS during follow-up in patients with AAV (Figure 3).

### 3.4. Cox Proportional Hazards Analysis

Similar to the ROC curve analysis results, in the univariable Cox analysis, serum AGP at AAV diagnosis exhibited no statistically significant association with all-cause mortality, relapse, ESKD, CVA, or ACS during follow-up in patients with AAV (Table 2).

## 4. Discussion

Given the clinical role of serum AGP as a biomarker with acute-phase reactant characteristics, this study investigated the usefulness of serum AGP measured at AAV diagnosis in reflecting the cross-sectional activity and predicting poor outcomes in immunosuppressive-drug-naïve patients with AAV. Finally, we obtained some interesting findings: The most important finding was that serum AGP at AAV diagnosis could reflect the cross-sectional activity of AAV and inflammatory burden represented by BVAS and ESR, respectively. However, we failed to demonstrate a significant correlation between serum AGP and other acute-phase reactants, including CRP or serum albumin measured at the same time point. Additionally, serum AGP at AAV diagnosis significantly correlated with the sum scores of the two items of the BVAS: ENT and pulmonary manifestations. Next, although serum AGP at diagnosis tended to be associated with ESKD occurrence during follow-up, serum AGP at AAV diagnosis showed no significant value in predicting the future occurrence of poor outcomes of AAV during follow-up. Therefore, we concluded that serum AGP at AAV diagnosis could potentially reflect cross-sectional activity in patients with AAV.

We wondered whether comparable to traditional acute-phase reactants such as ESR, CRP, serum albumin, and serum AGP could reflect or indicate the cross-sectional BVAS. First of all, we performed Pearson’s correlation analyses of the BVAS with ESR, CRP, and serum albumin at diagnosis, and found that the BVAS was significantly and remarkably correlated with ESR (r = 0.445, *p* < 0.001), CRP (r = 0.633, *p* < 0.001), and serum albumin (r = −0.714, *p* < 0.001) (Appendix A). To clarify the independent correlation power of serum AGP with the BVAS, we performed multivariable linear regression analysis of ESR, CRP, serum albumin, and serum AGP for cross-sectional BVAS at diagnosis. Among the four variables, serum AGP (standardised β = 0.177, *p* = 0.043), along with CRP (standardised β = 0.396, *p* = 0.001), and serum albumin (standardised β = −0.529, *p* < 0.001), was independently correlated with BVAS. Therefore, serum AGP was confirmed to have the potential to be an independent predictor of BVAS comparable to CRP and serum albumin.

AGP has been reported to possess both pro-inflammatory and anti-inflammatory mechanisms: for example, a low concentration of AGP could induce and accelerate lymphocyte proliferation and neutrophil aggregation, whereas a high concentration of AGP might have the opposite effect [12]. Particularly, as a pro-inflammatory inducer, AGP promotes the expression of inflammatory substances, such as interleukin (IL)-1 or IL-6, and these cytokines could subsequently enhance the expression of the downstream AGP gene as well [10]. Accordingly, when AGP plays an inflammatory role, AGP expression can be amplified, and conversely, when AGP shows an anti-inflammatory function, AGP expression can also be reduced. Therefore, the inflammation reflection range of serum AGP as an acute-phase reactant is very wide, so it can be said to have the advantage of being a highly sensitive biomarker for reflecting current AAV activity.

Meanwhile, among the nine systemic manifestation items of the BVAS [8], serum AGP was significantly correlated with the sum scores of ENT and pulmonary manifestations at AAV diagnosis but did not differ according to the presence of ENT or pulmonary manifestation. Nonetheless, unlike ENT manifestations, patients with pulmonary manifestation tended to exhibit elevated levels of serum AGP compared to patients without pulmonary manifestation. To date, no study has directly demonstrated the relationship between serum AGP and AAV-specific lung lesions, such as pulmonary fibrosis, interstitial lung disease, and lung nodule or cavitation. Rather, most previous studies revealed its potential as a biomarker in patients with lung cancers [14,15]. Therefore, we concluded that serum AGP contributes to the reflection of BVAS as a biomarker indicating the comprehensive amount of overall inflammation as an acute-phase reactant rather than directly showing AAV lung involvement.

Considering previous studies regarding the predictive potential of serum AGP for all-cause mortality and inflammatory risk factors for mortality in patients with AAV [16,17,18], we speculated that serum AGP may predict poor AAV outcomes. Firstly, in the ROC curve analysis of serum AGP for all-cause mortality, relapse, ESKD, CVA, and ACS, we found no statistical significance in AUC (Figure 3). Additionally, we examined the association between serum AGP and five poor outcomes using Cox proportional hazards analysis. However, no association of serum AGP with any poor outcome was observed (Table 2). Furthermore, although the predictive ability of serum AGP at diagnosis for ESKD occurrence during follow-up was not statistically significant, it was closer to statistical significance compared to other poor outcomes. Accordingly, using ROC curve analysis, the cut-off of serum AGP for ESKD occurrence was 1539.0 μg/mL (sensitivity 40.0%, and specificity 90.9%). In Kaplan–Meier survival analysis, patients with serum AGP ≥ 1539.0 μg/mL at AAV diagnosis exhibited a significantly reduced ESKD-free survival rate during follow-up than those with serum AGP < 1539.0 μg/mL (Appendix A). These results support the inference that serum AGP may reflect the accumulation of inflammation, which ultimately accelerates the progression to ESKD in patients with AAV, based on previous studies showing that serum AGP reflected the inflammatory burden of chronic kidney diseases [19,20].

This study’s strength lies in being the first to demonstrate the clinical utility of serum AGP in estimating the cross-sectional BVAS. Additionally, this study has other clinical implications as it suggests the potential to partially predict the occurrence of ESKD during follow-up in patients with AAV despite no statistical significance. This study has some limitations: First, the most critical limitation was the small sample size, as this was a pilot study exploring the clinical utility of serum AGP. Second, this study retrospectively assessed stored blood and clinical data, even though the study participants were selected from a prospective and observational cohort of Korean patients with AAV. Third, this study included only immunosuppressive-drug-naïve patients. Although it has the advantage of controlling for drug confounding variables, it also has the disadvantage of not reflecting patients taking drugs in real clinical practice. Fourth, because this study was designed as a pilot study to explore the possibility, it included no age- and sex-matched healthy controls. Thus, we could not provide reference levels of serum AGP or compare them between AAV patients and healthy controls. We expect that a prospective future study including not only more patients regardless of the administration of immunosuppressive drugs but also age- and sex-matched healthy controls will provide more reliable and realistic information on the clinical utility of serum AGP as a biomarker for assessing the current activity of AAV and predicting poor outcomes in patients with AAV.

## 5. Conclusions

In the present study, we demonstrated the clinical utility of serum AGP at AAV diagnosis in assessing the cross-sectional activity represented by the BVAS in patients with AAV for the first time. We expect that serum AGP will contribute to the assessment of the current activity of AAV as an additional biomarker in real clinical practice.

## Figures and Tables

**Figure 1 medicina-60-01212-f001:**
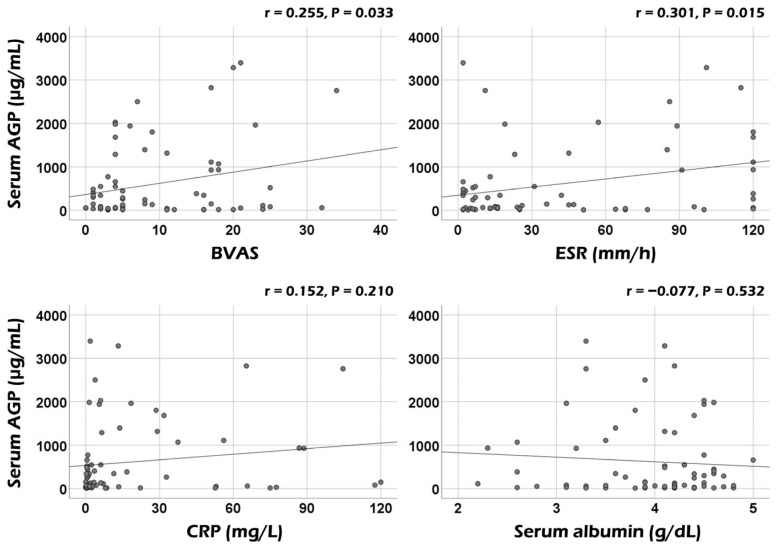
Correlation analyses of serum AGP with BVAS and acute-phase reactants. Serum AGP was significantly correlated with BVAS and ESR but not with CRP or serum albumin. AGP: alpha-1-acid glycoprotein; BVAS: Birmingham vasculitis activity score; ESR: erythrocyte sedimentation rate; CRP: C-reactive protein.

**Figure 2 medicina-60-01212-f002:**
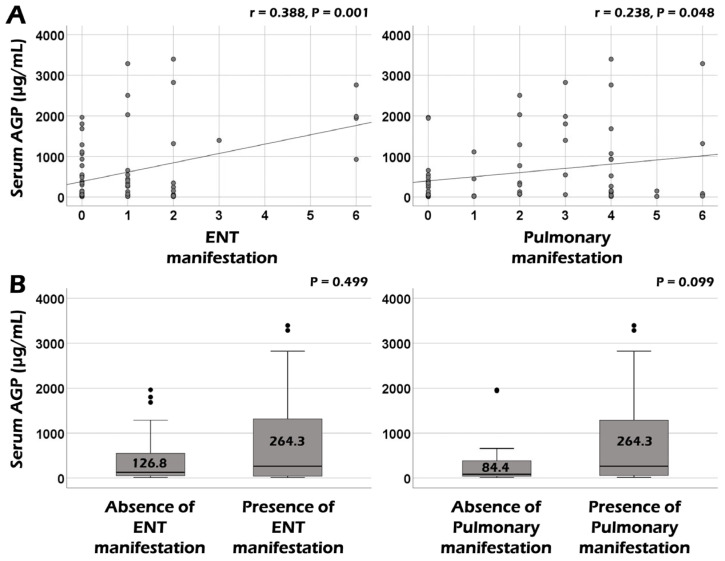
Correlation and comparative analyses of serum AGP according to BVAS items. (**A**) Serum AGP was significantly correlated with ENT and pulmonary manifestations among BVAS items. (**B**) Serum AGP was not significantly different between patients with and without ENT or pulmonary manifestations. The upper and lower error bars mean the maximum and minimum values, respectively. AGP: alpha-1-acid glycoprotein; ENT: ear–nose–throat.

**Figure 3 medicina-60-01212-f003:**
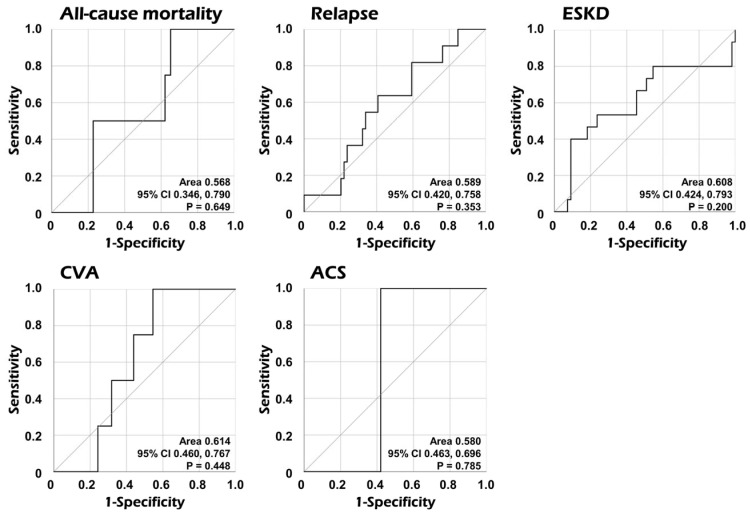
Comparative analyses of the area under the curve of serum AGP for each poor outcome. The AUC of serum AGP was not significant for each poor outcome. AGP: alpha-1-acid glycoprotein; CI: confidence interval; ESKD: end-stage kidney disease; CVA: cerebrovascular accident; ACS: acute coronary syndrome.

**Table 1 medicina-60-01212-t001:** Characteristics of patients with AAV at diagnosis and during follow-up (*N* = 70).

Variables	Values
At diagnosis	
Demographic data	
Age (years)	63.0 (52.0–72.3)
Male sex (*N*, (%))	29 (41.4)
Female sex (*N*, (%))	41 (58.6)
Ex-smoker (*N*, (%))	2 (2.9)
Body mass index (kg/m^2^)	224 (20.8–24.7)
AAV subtypes (*N*, (%))	
MPA	35 (50.0)
GPA	20 (28.6)
EGPA	15 (21.4)
ANCA titres and positivity (*N*, (%))	
MPO-ANCA (or P-ANCA) titre	0 (0–32.5)
PR3-ANCA (or C-ANCA) titre	0 (0–0)
MPO-ANCA (or P-ANCA)-positive	40 (57.1)
PR3-ANCA (or C-ANCA)-positive	11 (15.7)
Both ANCA-positive	2 (2.9)
ANCA-negative	21 (30.0)
AAV-specific indices	
BVAS	5.0 (3.0–17.0)
FFS	0 (0–1.0)
Comorbidities (*N*, (%))	
Type 2 diabetes mellitus	14 (20.0)
Hypertension	23 (32.9)
Dyslipidaemia	11 (15.7)
Acute-phase reactants	
ESR (mm/h)	23.0 (6.5–81.0)
CRP (mg/L)	3.4 (0.7–23.8)
Laboratory results	
White blood cell count (/mm^3^)	7495.0 (5930.0–10,485.0)
Haemoglobin (g/dL)	12.0 (10.2–13.6)
Platelet count (×1000/mm^3^)	248.0 (191.5–352.5)
Blood urea nitrogen (mg/dL)	18.7 (12.9–28.6)
Serum creatinine (mg/dL)	0.8 (0.6–1.6)
Total serum protein (g/dL)	6.8 (6.4–7.3)
Serum albumin (g/dL)	4.2 (3.6–4.4)
Serum AGP (μg/mL)	150.9 (41.6–929.6)
During follow-up	
Poor outcomes (*N*, (%))	
All-cause mortality	4 (5.7)
Relapse	11 (15.7)
ESKD	15 (21.4)
CVA	4 (5.7)
ACS	1 (1.4)
Follow-up duration based on each poor outcome (months)	
All-cause mortality	26.7 (12.3–45.9)
Relapse	22.4 (7.9–32.5)
ESKD	26.4 (9.2–45.9)
CVA	26.6 (11.9–43.1)
ACS	26.6 (12.1–43.1)
Medications (*N*, (%))	
Glucocorticoids	69 (98.6)
Cyclophosphamide	45 (64.3)
Rituximab	15 (21.4)
Mycophenolate mofetil	19 (27.1)
Azathioprine	42 (60.0)
Tacrolimus	6 (8.6)
Methotrexate	3 (4.3)

Values are expressed as a median (25–75 percentile) or *N* (%). ANCA: antineutrophil cytoplasmic antibody; AAV: ANCA-associated vasculitis; MPA: microscopic polyangiitis; GPA: granulomatosis with polyangiitis; MPO: myeloperoxidase; P: perinuclear; PR3: proteinase 3; C: cytoplasmic; BVAS: Birmingham vasculitis activity score; FFS: five-factor score; ESR: erythrocyte sedimentation rate; CRP: C-reactive protein; AGP: alpha-1-acid glycoprotein; ESKD: end-stage kidney disease; CVA: cerebrovascular accident; ACS: acute coronary syndrome.

**Table 2 medicina-60-01212-t002:** Cox proportional hazards analysis of serum AGP at diagnosis for poor outcomes during follow-up in patients with AAV.

Acute-Phase Reactants	Poor Outcomes	Univariable		
		HR	95% CI	*p* Value
Serum AGP	All-cause mortality	1.000	0.999, 1.001	0.840
	Relapse	1.000	1.000, 1.001	0.639
	ESKD	1.000	1.000, 1.001	0.175
	CVA	1.000	0.998, 1.001	0.741
	ACS	0.999	0.996, 1.003	0.759

AGP: alpha-1-acid glycoprotein; ANCA: antineutrophil cytoplasmic antibody; AAV: ANCA-associated vasculitis; ESKD: end-stage kidney disease; CVA: cerebrovascular accident; ACS: acute coronary syndrome.

## Data Availability

The dataset collected and/or analysed in the present study is available on request from the corresponding authors.

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
