# Peer review of "The Clinical Utility of Serum Alpha-1-Acid Glycoprotein in Reflecting the Cross-Sectional Activity of Antineutrophil Cytoplasmic Antibody-Associated Vasculitis: A Single-Centre Retrospective Study"

_medicina, 2024, doi:10.3390/medicina60081212_

Round 1

Reviewer 1 Report

Comments and Suggestions for Authors

In the manuscript entitled as: "Serum alpha-1-acid glycoprotein has the clinical utility in reflecting the cross-sectional activity of antineutrophil cytoplasmic antibody-associated vasculitis", Yoo et al, presented a potential clinical application of the serum AGP at antineutrophil cytoplasmic antibody-associated vasculitis (AAV) diagnosis. In addition, authors suggested the use of the serum AGP as an acute-phase marker to further predict poor outcomes during follow-up in patients with AAV.

The manuscript is well written and requires minor English language editing. Information provided are scientifically sound and of interest for the field. 

However, I would like to recommend a minor revision to improve the overall manuscript presentation. 

Introduction

- lines 40 - 41, please describe in more detail the clinical features of three subtypes of AAV.

-In line 43, please clarify what is defined under the cross-sectional activity and give a few examples on how the serum biomarkers (and which) could be predictive for cross-sectional activity of AAV. 

- Introduce here what you consider as the poor outcome for clarity (line 43).

- Rephrase the sentence in line 43 - 47. Separate the scoring-based indicators BVAS and FFS from markers such as ESR and CRP. Introduce and define properly BVAS and FFS for clarity.

- Start with a new paragraph  the sentence beginning with "..Alpha-1-acid glycoprotein (AGP).." in line 47.

Materials and Methods

- line 73, please list the medical conditions that can impair AAV diagnosis.

- Please add more information within this section, include the methodology of assessing the ANCA titer, techniques for ESR, CRP, and all other blood parameters listed in the Table1. 

Results 

- What are the normal serum AGP values in healthy adults? Inclusion of some control values would be suggested here. If this is not available, please discuss the normal serum AGP range in healthy controls and compare it with the levels reported in your manuscript. 

- Did you run correlation analysis between CRP and BVAS and ESR, and also between serum albumin and BVAS and ESR? If there would be no correlation, this would emphasize the potential application of serum AGP as the predictive marker.  This could be added as a supplementary figure. 

Discussion

- line 195: please rephrase the sentence, explain the proposed mechanism(s) of AGP anti-inlammatory action. 

-line 196: please rephrase. The sentence is too speculative and lack scientific soundness. 

- line 198: authors state that "..Numerous efforts have been made to discover serum biomarkers.." but then only one reference is provided. Please include more references or rephrase the statement. 

- line 201: which "items" of BVAS that are identified as indicators of the current AAV activity, are insensitive to real-time changes?

- from the line 202: This part is too broad. It speculates about serum biomarkers without defining them and without explaining the limitations including the issues with reliability and reproducibility. Also, the last sentence lines 205 - 207, please explain what was ment under the "serum AGP with a clear mechanism".

- lines 208 -214: please include references.

Comments on the Quality of English Language

Minor English editing required

Author Response

Reviewer (1)’s comments

Manuscript number: medicina-3092955

Title: Serum alpha-1-acid glycoprotein has the clinical utility in reflecting the cross-sectional activity of antineutrophil cytoplasmic antibody-associated vasculitis

We appreciate your excellent review of our manuscript. Your valuable comments helped us to make a better revision.

In the manuscript entitled as: "Serum alpha-1-acid glycoprotein has the clinical utility in reflecting the cross-sectional activity of antineutrophil cytoplasmic antibody-associated vasculitis", Yoo et al, presented a potential clinical application of the serum AGP at antineutrophil cytoplasmic antibody-associated vasculitis (AAV) diagnosis. In addition, authors suggested the use of the serum AGP as an acute-phase marker to further predict poor outcomes during follow-up in patients with AAV.

The manuscript is well written and requires minor English language editing. Information provided are scientifically sound and of interest for the field.

However, I would like to recommend a minor revision to improve the overall manuscript presentation.

Introduction

(1) lines 40 - 41, please describe in more detail the clinical features of three subtypes of AAV.

             As you recommended, we described in more detail the clinical features of three subtypes of AAV and amended the text in the INTRODUCTION section as below:

“AAV consists of three subtypes including microscopic polyangiitis (MPA), granulomatosis with polyangiitis (GPA), and eosinophilic GPA (EGPA) based on clinical features: MPA predominantly provokes vasculitis in the lungs and kidneys; GPA often affects the upper and lower respiratory tracts; and EGPA exhibits clinical manifestations related to both/either allergic and/or vasculitic components [3-6].” (Lines 40-44)

(2) In line 43, please clarify what is defined under the cross-sectional activity and give a few examples on how the serum biomarkers (and which) could be predictive for cross-sectional activity of AAV.

             According to your indication, we came to know that these sentences may be misleading because the following sentence mentions the advantages of serum biomarkers over BVAS.

In real clinical settings, BVAS is the most widely used index to assess the current activity of AAV. Accordingly, to avoid the confusion, we amended these sentences as below:

 “To date, besides conventional inflammatory markers including erythrocyte sedimentation rate (ESR), and C-reactive protein (CRP), numerous efforts have been made to identify various serum biomarkers that can not only estimate and reflect the cross-sectional activity of AAV assessed by the Birmingham vasculitis activity score (BVAS) but also anticipate the poor outcomes predicted by the five-factor score (FFS) such as all-cause mortality or kidney function impairment. In particular, if BVAS could be replaced with newly developed serum biomarkers positively, they may have advantages, such as convenience, reproducibility-reliability, and dynamic information compared to traditional AAV-specific indices [7-9].” (Lines 44-53)

(3) Introduce here what you consider as the poor outcome for clarity (line 43).

             As you recommended, we added the examples of poor outcomes of AAV in the text in the INTRODUCTION section as below:

“as well as the poor outcomes of AAV, for instance, all-cause mortality or kidney function impairment requiring renal replacement therapy, predicted by the five-factor score (FFS).” (Lines 48-50)

(4) Rephrase the sentence in line 43 - 47. Separate the scoring-based indicators BVAS and FFS from markers such as ESR and CRP. Introduce and define properly BVAS and FFS for clarity.

             According to your comment, we provided the clinical roles of BVAS and FFS and amended the paragraph in the INTRODUCTION section like the responses to comments 2 and 3.

“To date, besides conventional inflammatory markers including erythrocyte sedimentation rate (ESR), and C-reactive protein (CRP), numerous efforts have been made to identify various serum biomarkers that can not only estimate and reflect the cross-sectional activity of AAV assessed by the Birmingham vasculitis activity score (BVAS) but also anticipate the poor outcomes predicted by the five-factor score (FFS) such as all-cause mortality or kidney function impairment. In particular, if BVAS could be replaced with newly developed serum biomarkers positively, they may have advantages, such as convenience, reproducibility-reliability, and dynamic information compared to traditional AAV-specific indices [7-9].” (Lines 44-53)

(5) Start with a new paragraph the sentence beginning with "..Alpha-1-acid glycoprotein (AGP).." in line 47.

             We started a new paragraph with the sentence beginning with “Alpha-1-acid glycoprotein (AGP) is a negatively charged glycosylated single-chain protein, mainly synthesised, and catalysed in the liver [10].” (Lines 54-55)

Materials and Methods

(6) line 73, please list the medical conditions that can impair AAV diagnosis.

             According to your recommendation, we added the concrete cases of serious medical conditions that can impair AAV diagnosis to the text in the METHODS section as below:

“The exclusion criteria were: i) patients who had serious medical conditions such as concomitant malignancies, and severe infectious diseases requiring hospitalisation at AAV diagnosis; ii) patients who were diagnosed with overlap syndrome with other types of systemic vasculitis at AAV diagnosis; and ii) patients who were exposed to immunosuppressive drugs for AAV treatment within four weeks before AAV diagnosis.” (Lines 83-87)

(7) Please add more information within this section, include the methodology of assessing the ANCA titre, techniques for ESR, CRP, and all other blood parameters listed in the Table1.

             As you recommended, we added the ANCA tests methodology, and the list of routinely performed laboratory tests including ESR and CRP to the text in the METHODS section as below:

“Perinuclear (P)-ANCA and cytoplasmic (C)-ANCA were detected by an indirect immuno-fluorescence assay, whereas, myeloperoxidase (MPO)-ANCA and proteinase 3 (PR3)-ANCA were measured by an immunoassay, the novel anchor coated highly sensitive Phadia EliA (Thermo Fisher Scientific/Phadia, Freiburg, Germany) using human native antigens, performed on a Phadia 250 analyser. According to the 2022 ACR/EULAR criteria for AAV, in the present study, both MPO-ANCA/PR3-ANCA and P-ANCA/C-ANCA were all recognised as ANCA results [3-5]. Comorbidities included type 2 diabetes mellitus, hypertension, and dyslipidaemia. Acute-phase reactants including ESR and CRP (reference range from 0 to 8 mg/L) were recorded. The results of routinely performed laboratory tests such as white blood cell count, haemoglobin, platelet count, blood urea nitrogen, serum creatinine, total serum protein, and serum albumin, were also collected.” (Lines 92-103)

Results

(8) What are the normal serum AGP values in healthy adults? Inclusion of some control values would be suggested here. If this is not available, please discuss the normal serum AGP range in healthy controls and compare it with the levels reported in your manuscript.

             Because this study was designed as a pilot study to explore the possibility, the present study included no age- and sex-matched healthy controls. Thus, we could not provide reference levels of serum AGP and compare them between AAV patients and healthy controls. So, we attempted to search the literature for serum AGP values of healthy controls but did not proceed because we thought that it would not be of significance unless they will be data from age- and sex-matched healthy controls taking account into ethnic and geographical differences.

             If you do not mind, we would like to add these contents to the text in the LIMITATIONS section as below:

“Fourth, because this study was designed as a pilot study to explore the possibility, the present study included no age- and sex-matched healthy controls. Thus, we could not provide reference levels of serum AGP and compare them between AAV patients and healthy controls. We expect that a prospective future study including not only more patients regardless of the administration of immunosuppressive drugs but also age- and sex-matched healthy controls will provide more reliable and realistic information on the clinical utility of serum AGP as a biomarker for assessing the current activity of AAV and predicting poor outcomes in patients with AAV.” (Lines 271-279)

(9) Did you run correlation analysis between CRP and BVAS and ESR, and also between serum albumin and BVAS and ESR? If there would be no correlation, this would emphasize the potential application of serum AGP as the predictive marker.  This could be added as a supplementary figure.

             As you recommended, we performed Pearson’s correlation analyses of BVAS with ESR, CRP, and serum albumin at diagnosis, and found that BVAS was significantly correlated with ESR (r = 0.445, P <0.001), CRP (r = 0.633, P <0.001), and serum albumin (r = −0.714, P <0.001). Additionally, to clarify the independent correlation power of serum AGP with BVAS, we performed multivariable linear regression analysis of ESR, CRP, serum albumin, and serum AGP for cross-sectional BVAS at diagnosis. Among the four variables, serum AGP (standardised β = 0.177, P = 0.043), along with CRP (standardised β = 0.396, P = 0.001), and serum albumin (standardised β = −0.529, P <0.001), was independently correlated with BVAS. Therefore, serum AGP was confirmed to have the potential to be an independent predictor of BVAS comparable to CRP and serum albumin.

             We added Supplementary Figure 0 to the manuscript, and also added the contents of the supplementary file to the text in the DISCUSSION section as below:

Supplementary Materials: The following supporting information can be downloaded at: www.mdpi.com/xxx/s1, Figure S1: Correlation of BVAS with ESR, CRP, and serum albumin at diagnosis; Figure S2: Comparison of cumulative ESKD-free survival rates. (Lines 285-287)

“We wondered whether comparable to traditional acute-phase reactants such as ESR, CRP, and serum albumin, serum AGP could reflect or guess cross-sectional BVAS. First of all, we performed Pearson’s correlation analyses of BVAS with ESR, CRP, and serum albumin at diagnosis, and found that BVAS was significantly and remarkably correlated with ESR (r = 0.445, P <0.001), CRP (r = 0.633, P <0.001), and serum albumin (r = −0.714, P <0.001) (Figure S1). To clarify the independent correlation power of serum AGP with BVAS, we performed multivariable linear regression analysis of ESR, CRP, serum albumin, and serum AGP for cross-sectional BVAS at diagnosis. Among the four variables, serum AGP (standardised β = 0.177, P = 0.043), along with CRP (standardised β = 0.396, P = 0.001), and serum albumin (standardised β = −0.529, P <0.001), was independently correlated with BVAS. Therefore, serum AGP was confirmed to have the potential to be an independent predictor of BVAS comparable to CRP and serum albumin.” (Lines 208-219)

Discussion

(10) line 195: please rephrase the sentence, explain the proposed mechanism(s) of AGP anti-inflammatory action.

(11) line 196: please rephrase. The sentence is too speculative and lack scientific soundness.

We respond to both comments 10 and 11 as well. 

Thanks to your comment, we could find a logical error. What we wanted to talk about is as follows: First of all, AGP has been reported to possess both pro-inflammatory and anti-inflammatory mechanisms: for example, a low concentration of AGP could induce and accelerate lymphocyte proliferation and neutrophil aggregation, whereas, a high concentration of AGP might have the opposite effect [12]. Particularly, as a pro-inflammatory inducer, AGP promotes the expression of inflammatory substances, such as interleukin (IL)-1 or IL-6, and these cytokines could subsequently enhance the expression of the downstream AGP gene as well [10]. Accordingly, when AGP plays an inflammatory role, AGP expression can be amplified, and conversely, when AGP shows an anti-inflammatory function, AGP expression can also be reduced. Therefore, the inflammation reflection range of serum AGP as an acute-phase reactant is very wide, so it can be said to have the advantage of being a highly sensitive biomarker for reflecting current AAV activity.

We replaced the pre-existing paragraph with these contexts and rephrased the sentences in the DISCUSSION section as below:

“AGP has been reported to possess both pro-inflammatory and anti-inflammatory mechanisms: for example, a low concentration of AGP could induce and accelerate lymphocyte proliferation and neutrophil aggregation, whereas, a high concentration of AGP might have the opposite effect [12]. Particularly, as a pro-inflammatory inducer, AGP promotes the expression of inflammatory substances, such as interleukin (IL)-1 or IL-6, and these cytokines could subsequently enhance the expression of the downstream AGP gene as well [10]. Accordingly, when AGP plays an inflammatory role, AGP expression can be amplified, and conversely, when AGP shows an anti-inflammatory function, AGP expression can also be reduced. Therefore, the inflammation reflection range of serum AGP as an acute-phase reactant is very wide, so it can be said to have the advantage of being a highly sensitive biomarker for reflecting current AAV activity.” (Lines 220-230)

(12) - line 198: authors state that "..Numerous efforts have been made to discover serum biomarkers.." but then only one reference is provided. Please include more references or rephrase the statement.

(13) line 201: which "items" of BVAS that are identified as indicators of the current AAV activity, are insensitive to real-time changes?

(14) from the line 202: This part is too broad. It speculates about serum biomarkers without defining them and without explaining the limitations including the issues with reliability and reproducibility. Also, the last sentence lines 205 - 207, please explain what was ment under the "serum AGP with a clear mechanism".

We respond to the three comments 12, 13, and 14. 

             We believe that we have sufficiently emphasized the merits of serum AGP's AAV activity reflection in the new paragraph above. In addition, we believe that the content of paragraphs 198-207 may cause contradictory misunderstandings rather than positive contributions to this paper. Therefore, to avoid confusion in the meaning, we thought that it would be appropriate to delete the content of lines 198-207 from this paper and thus, we removed them.

Numerous efforts have been made to discover serum biomarkers that can indicate the cross-sectional activity or predict poor outcomes in patients with AAV [7]. Tradition-ally, among the items of BVAS as indicators of the current AAV activity, some are insensitive to real-time changes. Therefore, their reflection of the current activity remains unclear. Moreover, they have limitations as they are not practical for a repeated assessment every few days. However, the discovery of such serum biomarkers is expected to overcome these limitations and increase convenience, reliability, and reproducibility. Regarding this need, serum AGP with a clear mechanism has clinical significance as a useful inflammation-related biomarker.

(15) lines 208 -214: please include references.

             According to your recommendation, we added the reference number and amended the text in the DISCUSSION section as below:

“Meanwhile, among nine systemic manifestation items of BVAS [8],” (Lines 231)

             Additionally, to avoid unnecessary and redundant mentions, we removed the sentence in the DISCUSSION section as below:

Previous studies have scarcely demonstrated an association between serum AGP and pulmonary manifestations of AAV.

“Nonetheless, unlike ENT manifestations, patients with pulmonary manifestation tended to exhibit elevated levels of serum AGP compared to patients without pulmonary manifestation. To date, no study has directly demonstrated the relationship between serum AGP and AAV-specific lung lesions, such as pulmonary fibrosis, interstitial lung disease, and lung nodule or cavitation. Rather, most previous studies revealed its potential as a biomarker in patients with lung cancers [14, 15].” (Lines 234-239)

Reviewer 2 Report

Comments and Suggestions for Authors

The manuscript is a pilot project for matching cross-sectional activity of antineutrophil cytoplas-3 mic antibody-associated vasculitis. The article has pleasant structure, main idea and results are described clearly. 

I recommend to introduce some minor changes (clarifications).

1. p.1 lines 43-47. "Serum 43 biomarkers have advantages.... and C-reactive protein (CRP) [7-9]."  Please specify which proved biomarkers are exist for this clinical situation.

2. p. 7, lines 213-214. "Previous studies have scarcely demonstrated an association between serum AGP and 213 pulmonary manifestations of AAV." Please sign the reference.

Author Response

Reviewer (2)’s comments

Manuscript number: medicina-3092955

Title: Serum alpha-1-acid glycoprotein has the clinical utility in reflecting the cross-sectional activity of antineutrophil cytoplasmic antibody-associated vasculitis

We appreciate your excellent review of our manuscript. Your valuable comments helped us to make a better revision.

The manuscript is a pilot project for matching cross-sectional activity of antineutrophil cytoplas-3 antibody-associated vasculitis. The article has pleasant structure, main idea and results are described clearly.

I recommend to introduce some minor changes (clarifications).

(1) p.1 lines 43-47. "Serum 43 biomarkers have advantages.... and C-reactive protein (CRP) [7-9]."  Please specify which proved biomarkers are exist for this clinical situation.

             We agree with your indication. To clarify the meaning, we amended the text in the INTRODUCTION section as below:

“To date, besides conventional inflammatory markers including erythrocyte sedimentation rate (ESR), and C-reactive protein (CRP), numerous efforts have been made to identify various serum biomarkers that can simultaneously estimate and reflect the cross-sectional activity of AAV assessed by the Birmingham vasculitis activity score (BVAS) as well as the poor outcomes of AAV, for instance, all-cause mortality or kidney function impairment requiring renal replacement therapy, predicted by the five-factor score (FFS). In particular, if BVAS could be replaced with newly developed serum biomarkers positively, they may have advantages, such as convenience, reproducibility-reliability, and dynamic information compared to traditional AAV-specific indices [7-9].” (Lines 44-53)

(2) p. 7, lines 213-214. "Previous studies have scarcely demonstrated an association between serum AGP and 213 pulmonary manifestations of AAV." Please sign the reference.

             We recognised that the sentence you mentioned and the next sentence are duplicates of the same content, and the next sentence contains more clear content. Therefore, we decided that it was appropriate to delete the sentence you mentioned, and we removed it in the DISCUSSION section as below:

Previous studies have scarcely demonstrated an association between serum AGP and pulmonary manifestations of AAV.

“Nonetheless, unlike ENT manifestations, patients with pulmonary manifestation tended to exhibit elevated levels of serum AGP compared to patients without pulmonary manifestation. To date, no study has directly demonstrated the relationship between serum AGP and AAV-specific lung lesions, such as pulmonary fibrosis, interstitial lung disease, and lung nodule or cavitation. Rather, most previous studies revealed its potential as a biomarker in patients with lung cancers [14, 15].” (Lines 234-239)

Reviewer 3 Report

Comments and Suggestions for Authors

I appreciate the opportunity to review the manuscript entitled "Serum alpha-1-acid glycoprotein has the clinical utility in reflecting the cross-sectional activity of antineutrophil cytoplasmic antibody-associated vasculitis". The research question is scientifically valid. However, The methodology is ambiguous and needs more details in chronological order. The authors should take note of the major remarks listed below to improve the manuscript:

Major comments

1. The authors should explain in detail the exclusion criteria, the duration of the recruitment period, and how these subjects were selected. Please cite reference no. 7 here, if relevant. The authors can follow STROBE guidelines.

2. Line no. 73: Please list the names of serious conditions that should be absent in the selected patients.

3. Please add the study type (retrospective study) in the methods section. Was the clinical data collected from the hospital's electronic system? If so, please mention that.

4. In bar graphs, what do error bars represent? Please mention.

5. Please provide references to the line no. 200 (TRaditionally, among...) and 213 (Previous studies...)

6. Is it possible to check the correlation between serum AGP and delta neutrophil index (DNI)? Moreover, would CRP/serum albumin ratio (CAR) show any relation?

Minor comments

1. Please mention the version of BVAS.

Comments on the Quality of English Language

A few corrections are needed. For example,

1. line no. 112: respective

2. line no. 118: For consistency, "15" should be spelled out.

Author Response

Reviewer (3)’s comments

Manuscript number: medicina-3092955

Title: Serum alpha-1-acid glycoprotein has the clinical utility in reflecting the cross-sectional activity of antineutrophil cytoplasmic antibody-associated vasculitis

We appreciate your excellent review of our manuscript. Your valuable comments helped us to make a better revision.

I appreciate the opportunity to review the manuscript entitled "Serum alpha-1-acid glycoprotein has the clinical utility in reflecting the cross-sectional activity of antineutrophil cytoplasmic antibody-associated vasculitis". The research question is scientifically valid. However, The methodology is ambiguous and needs more details in chronological order. The authors should take note of the major remarks listed below to improve the manuscript:

Major comments

(1) The authors should explain in detail the exclusion criteria, the duration of the recruitment period, and how these subjects were selected. Please cite reference no. 7 here, if relevant. The authors can follow STROBE guidelines.

             As you indicated, we amended the text regarding the exclusion criteria, the duration of the recruitment period, and patients’ selection in the METHODS section as below:

“The exclusion criteria were: i) patients who had serious medical conditions such as concomitant malignancies, and severe infectious diseases requiring hospitalisation at AAV diagnosis; ii) patients who were diagnosed with overlap syndrome with other types of systemic vasculitis at AAV diagnosis; and ii) patients who were exposed to immunosuppressive drugs for AAV treatment within four weeks before AAV diagnosis.” (Lines 83-87)

“In the present study, we randomly selected 70 patients from the Severance Hospital ANCA-associated VasculitidEs (SHAVE) cohort, an observational cohort of Korean pa-tients with AAV from November 2005. The electronic medical records of the 70 patients were retrospectively reviewed.” (Lines 70-73)

             Whereas, reference number 7 is a review article summarising the currently developed and investigating serological biomarkers in AAV patients. Therefore, if you do not mind, it would be better not to cite it here.

(2) Line no. 73: Please list the names of serious conditions that should be absent in the selected patients.

             As we described above, we listed the names of serious conditions defined in this study in the METHODS section as below:

“The exclusion criteria were: i) patients who had serious medical conditions such as concomitant malignancies, and severe infectious diseases requiring hospitalisation at AAV diagnosis;” (Lines 83-85)

(3) Please add the study type (retrospective study) in the methods section. Was the clinical data collected from the hospital's electronic system? If so, please mention that.

“In the present study, we randomly selected 70 patients from the Severance Hospital ANCA-associated VasculitidEs (SHAVE) cohort, an observational cohort of Korean patients with AAV from November 2005. The electronic medical records of the 70 patients were retrospectively reviewed.” (Lines 70-73)

(4) In bar graphs, what do error bars represent? Please mention.

             According to your indication, we added the explanation what errors bars represented to the FIGURE 2 LEGEND as below:

“The upper and lower error bars mean the maximum and minimum values, respectively.” (Line 169)

(5) Please provide references to the line no. 200 (Traditionally, among...) and 213 (Previous studies...)

             We appreciate your comment and agree with the need for references in this section. However, in response to another reviewer's comment, the two parts that you pointed out that required references were both removed. So, I will describe the details here again.

             First, in terms of the contents in line 200, we replaced the pre-existing paragraph with these contexts and rephrased the sentences in the DISCUSSION section as below:

“AGP has been reported to possess both pro-inflammatory and anti-inflammatory mechanisms: for example, a low concentration of AGP could induce and accelerate lymphocyte proliferation and neutrophil aggregation, whereas, a high concentration of AGP might have the opposite effect [12]. Particularly, as a pro-inflammatory inducer, AGP promotes the expression of inflammatory substances, such as interleukin (IL)-1 or IL-6, and these cytokines could subsequently enhance the expression of the downstream AGP gene as well [10]. Accordingly, when AGP plays an inflammatory role, AGP expression can be amplified, and conversely, when AGP shows an anti-inflammatory function, AGP expression can also be reduced. Therefore, the inflammation reflection range of serum AGP as an acute-phase reactant is very wide, so it can be said to have the advantage of being a highly sensitive biomarker for reflecting current AAV activity.” (Lines 220-230)

             And, we believe that we have sufficiently emphasized the merits of serum AGP's AAV activity reflection in the new paragraph above. In addition, we believe that the content of paragraphs 198-207 may cause contradictory misunderstandings rather than positive contributions to this paper. Therefore, to avoid confusion in the meaning, we thought that it would be appropriate to delete the content of lines 198-207 from this paper and thus, we removed them.

Numerous efforts have been made to discover serum biomarkers that can indicate the cross-sectional activity or predict poor outcomes in patients with AAV [7]. Tradition-ally, among the items of BVAS as indicators of the current AAV activity, some are insensitive to real-time changes. Therefore, their reflection of the current activity remains unclear. Moreover, they have limitations as they are not practical for a repeated assessment every few days. However, the discovery of such serum biomarkers is expected to overcome these limitations and increase convenience, reliability, and reproducibility. Regarding this need, serum AGP with a clear mechanism has clinical significance as a useful inflammation-related biomarker.

             Second, in terms of the contents in line 213, we recognised that the sentence you mentioned and the next sentence are duplicates of the same content, and the next sentence contains more clear content. Therefore, we decided that it was appropriate to delete the sentence you mentioned, and we removed it in the DISCUSSION section as below:

Previous studies have scarcely demonstrated an association between serum AGP and pulmonary manifestations of AAV.

(6) Is it possible to check the correlation between serum AGP and delta neutrophil index (DNI)? Moreover, would CRP/serum albumin ratio (CAR) show any relation?

             First, regarding DNI, the patients included in this study did not have DNI values measured at the same time point as serum AGP measurement. Therefore, we could not perform the correlation analysis between serum AGP and DNI at diagnosis. Second, regarding CRP and serum albumin ratio (CAR), in this study, no significant correlation between serum AGP and CAR measured and calculated at diagnosis was observed.

Minor comments

(1) Please mention the version of BVAS.

             According to your recommendation, we added the version of BVAS to the text in the METHODS section as below:

“AAV subtype, ANCA positivity, and AAV-specific indices including, BVAS (version 3) [8], and FFS [9], were reviewed.” (Lines 91-5-92)

Additional comments

Comments on the Quality of English Language

A few corrections are needed. For example,

We revised the manuscript according to English editing.
